# Earth Observation-Based Detectability of the Effects of Land Management Programmes to Counter Land Degradation: A Case Study from the Highlands of the Ethiopian Plateau

**Esther Barvels * and Rasmus Fensholt** 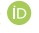

Department of Geosciences and Natural Resource Management, University of Copenhagen,
DK-1350 Copenhagen, Denmark; rf@ign.ku.dk
* Correspondence: lbm692@alumni.ku.dk

**Abstract:** In Ethiopia land degradation through soil erosion is of major concern. Land degradation mainly results from heavy rainfall events and droughts and is associated with a loss of vegetation and a reduction in soil fertility. To counteract land degradation in Ethiopia, initiatives such as the Sustainable Land Management Programme (SLMP) have been implemented. As vegetation condition is a key indicator of land degradation, this study used satellite remote sensing spatiotemporal trend analysis to examine patterns of vegetation between 2002 and 2018 in degraded land areas and studied the associated climate-related and human-induced factors, potentially through interventions of the SLMP. Due to the heterogeneity of the landscapes of the highlands of the Ethiopian Plateau and the small spatial scale at which human-induced changes take place, this study explored the value of using 30 m resolution Landsat data as the basis for time series analysis. The analysis combined Landsat derived Normalised Difference Vegetation Index (NDVI) data with Climate Hazards group Infrared Precipitation with Stations (CHIRPS) derived rainfall estimates and used Theil-Sen regression, Mann-Kendall trend test and LandTrendr to detect changes in NDVI, rainfall and rain-use efficiency. Ordinary Least Squares (OLS) regression analysis was used to relate changes in vegetation directly to SLMP infrastructure. The key findings of the study are a general trend shift from browning between 2002 and 2010 to greening between 2011 and 2018 along with an overall greening trend between 2002 and 2018. Significant improvements in vegetation condition due to human interventions were found only at a small scale, mainly on degraded hillside locations, along streams or in areas affected by gully erosion. Visual inspections (based on Google Earth) and OLS regression results provide evidence that these can partly be attributed to SLMP interventions. Even from the use of detailed Landsat time series analysis, this study underlines the challenge and limitations to remotely sensed detection of changes in vegetation condition caused by land management interventions aiming at countering land degradation.

**Keywords:** developing countries; Google Earth Engine; land degradation; Landsat time series analysis; semi-arid areas; sustainable land management programmes

## 1. Introduction

Degradation of land and soil affects approximately one third of the global land area that is used for agriculture [1], involving livelihoods of more than 1.5 million people [2]. Land degradation is of particular concern in developing countries, as this issue poses a threat to food security for a large number of poor people and to local economic activities [3]. In the United Nation's Convention to Combat Desertification (UNCCD) (art. 1f) land degradation is defined as "reduction or loss, in arid, semi-arid and dry sub-humid areas, of the biological or economic productivity and complexity of rainfed cropland, irrigated cropland, or range, pasture, forest and woodlands resulting from land uses or from a process or combination of processes, including processes arising from human activities and habitation patterns". Among those processes are soil erosion or long-term loss of

natural vegetation [4]. In Ethiopia, land degradation results mainly from soil erosion by water [5] and occurs particularly in the Ethiopian highlands which are inhabited by 88% of the national population, cover 60% of the national livestock resources and encompass 90% of the area suitable for agriculture [6]. The country's natural physical conditions are one of the underlying drivers of land degradation. Ethiopia has always been prone to soil erosion and droughts due to high rainfall variability, which causes reduced vegetation cover in dry years and soil loss in subsequent wet years [7]. In Ethiopia's highlands, soil erosion is facilitated by steep terrain with slopes in excess of 30% [5]. Moreover, population pressure, increasing livestock (and with it deforestation), overgrazing (due to uncontrolled free grazing) and the expansion of agricultural fields into marginal land are underlying drivers of soil erosion [5,8]. The shortage of fertile cropland led to a shifting of cattle and livestock grazing activities to areas that are specifically vulnerable to soil erosion such as deforested, ecologically fragile hillsides with steep slopes. Gully formation, the removal of soil along drainage lines (channels) by surface water runoff, is one of the apparent consequences of soil erosion in Ethiopia [8].

The restoration of degraded land and soil, the implementation of sustainable land management (SLM) and resilient agricultural practices are targeted in the United Nation's Sustainable Development Goals (SDG). To address in particular target 15.3 which aims to combat desertification and restore degraded and soil, UNCCD adopted the Land Degradation Neutrality (LDN) Target Setting Programme. Achieving LDN will also contribute to reaching other SDGs including those on poverty reduction, food and water [9]. To counter desertification and land degradation UNCCD and affected developing countries have set voluntary targets and implemented National Action Programmes that are supported by international cooperation, including financial and technical resources [4]. In this context, developing countries have implemented land management and land restoration projects on both national and local levels in collaboration with multilateral and bilateral development partners. To rehabilitate degraded landscapes and scale up SLM in Ethiopia, the Ethiopian government launched the Sustainable Land Management Programme (SLMP) in 2009 in collaboration with a range of international donors including the World Bank and the German Development Bank (KfW) [10]. The impact of Sustainable Land Management (SLM) in Ethiopia has been studied through runoff and soil loss measurements [11] and analyses within the economics field, for instance by examining household data to assess the effect on crop yields [12]. A study by Ali et al. used earth observation derived vegetation indices to estimate the impact of SLM in a single watershed in Ethiopia [13].

Monitoring of land surface dynamics, such as land degradation ('browning') or land recovery ('greening') is widely done by implementing earth observation time series analysis. A plethora of studies have examined long term trends by applying ordinary least square (OLS) linear regression models, e.g., regressing vegetation indices with time, based on high temporal resolution data such as from Advanced Very High Resolution Radiometer onboard National Oceanic and Atmospheric Administration (AVHRR-NOAA) or from Moderate Resolution Imaging Spectroradiometer (MODIS). For the Sahel, linear trends of yearly NDVI anomalies derived from AVHRR [14] or linear trends of the seasonal NDVI amplitude and integral [15] have been examined to monitor vegetation. MODIS NDVI data have been used to detect land degradation and regeneration processes in the Sahel [16], Mongolia [17] and Ethiopia [18].

Studies of semi-arid areas [19–22] and of Ethiopia [23] have demonstrated a strong relationship between rainfall and NDVI. Therefore, methods have been developed to disentangle rainfall-related effects from human-induced effects on land degradation. The rain-use efficiency (RUE), defined as the ratio of above-ground net primary production (ANPP) to annual precipitation [24], has been used to detect non-precipitation related land degradation. The basic assumption involved in the use of RUE is the existence of a constant linear relationship between vegetation productivity (or ANPP) and precipitation in areas where land is not affected by human-induced degradation [25]. By normalising for the effect of interannual rainfall variability on ANPP, human-induced changes can then be singled

out [26]. RUE has been used as a measure in several studies, e.g., of the Sahel [25,27], South Africa [19], global drylands [2] and Northern Eurasia [26]. Furthermore, residual trend analysis, i.e., analysing the residuals from a NDVI-rainfall regression model, was found to be effective in disentangling the climate effects from human-induced land degradation [28,29]. In areas with steep terrain land degradation can be driven by climate effects such as high rainfall variability. Hermans-Neumann et al. combined NPP trends, precipitation variability and census data to identify areas in Ethiopia where high in-migration is coupled with land degradation, proposing the latter is likely occurred due to human activities [18].

Since the opening of the Landsat archive by the United States Geological Survey (USGS) in 2008, an increasing amount of studies that exploit medium/high resolution data for time series analysis has been published [30,31]. In parallel, new change detection methods that not only account for linear (in this case gradual trends), but also for abrupt occurrences by separating time series into individual segments, have been developed. Amongst those are Landsat-based detection of Trends in Disturbance and Recovery (LandTrendr) and Breaks For Additive Seasonal and Trend (BFAST) which have been widely used for vegetation monitoring. BFAST has for example been used to detect gradual and abrupt changes in NDVI and rain-use efficiency [26,32–34] and water-use efficiency [35]. LandTrendr has been widely used for monitoring forest disturbances (fire or stand clearing) and forest regrowth [36–38], forest biomass [39] and for agricultural and land abandonment mapping [40].

The aim of this study was to temporally and spatially analyse vegetation dynamics in degraded land areas in Ethiopia between 2002 and 2018 in relation to management programmes implemented to counter land degradation. Due the heterogeneity of the landscapes of the areas examined in this study and the fact that human-induced changes are expected to take place at a small spatial scale, this study aimed at exploring the value of using medium resolution Landsat data derived from different sensors for trend analysis at a spatial and temporal scale compatible with the scale of SLMP interventions. The investigated areas had gone through interventions aiming at avoiding further land degradation and increasing vegetation cover. In this context, associated human-induced and climate-related factors of land degradation and land recovery were examined. The specific objectives of this study were:

1. The examination of spatiotemporal vegetation trends using Landsat time series and to analyse their forcing mechanisms (climate-related vs. human-induced).
2. The assessment of the detectability of the impact from typical SLMP interventions on vegetation conditions from the use of relevant remote sensing data sources available at no costs.

## 2. Materials and Methods

### 2.1. Study Area

The study area consists of 21 major watersheds which are distributed in three different zones of Ethiopia, in Amhara, Oromia and Tigray, and have mean altitudes between 1200 and 3100 m (Figure 1), mean slopes up to 14.4 degrees and mean annual rainfall between approx. 600 and 1900 mm per watershed (calculations based on CHIRPS data). The watersheds are located in semi-arid and sub-humid agro-ecological zones where temperate to cool climate prevails and are surrounded by low-lying tropical warm to hot savannas and semidesert regions [18,41]. They are characterised by heterogeneous landscapes, with croplands and grasslands as the most dominant land cover and hillsides, which have been degraded and closed for farming and grazing. Agriculture is dominated by small-scale subsistence mixed farming systems, i.e., crop production mixed with livestock rearing activities [41]. Crops are mainly grown during the wet period from March through September with harvest taking place mostly in October to December [18].

The watersheds constitute intervention areas of the Sustainable Land Management Programme, a multi-donor supported project, first implemented by the Government of Ethiopia in 2009 and phased out in 2019. SLMP's overall goal was to "reduce land

degradation and improve land productivity in selected watersheds in targeted regions in Ethiopia" [10]. It's first component, watershed and landscape management, aimed at reforesting and afforesting degraded communal land, increasing agricultural and livestock productivity, reducing carbon emission, building climate resilience and increasing water availability. To achieve these goals, activities such as hillside communal land treatment, including the prohibition of free grazing, gully rehabilitation and cropland treatment using biophysical measures, promoting agro-forestry and fodder production, and the construction of water harvesting structures were supported in the watersheds [10].

The 21 major watersheds each comprise between 6 to 20 micro-watersheds (Figure 1) with a total of 314 micro-watersheds and an average area of 7 km$^2$. 220 micro-watersheds (1541 km$^2$) received SLMP support from 2011 to 2019 by KfW with the technical assistance of the German Agency for International Coorporation (GIZ). In the following, the supported micro-watersheds are referred to as treatment areas while the remaining 94 micro-watersheds (543 km$^2$) that did not receive any support are referred to as control areas.

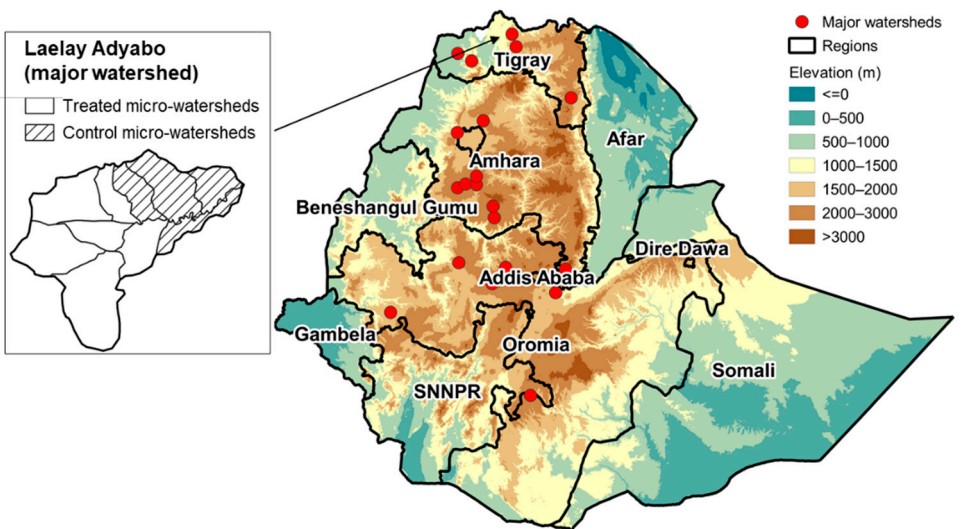

**Figure 1.** Overview map of the location of the study areas (i.e., major watersheds) and elevation.

### 2.2. Data

Landsat Collection 1 atmospherically corrected Surface Reflectance (SR) Tier 1 products were used for the period 2001–2019 including three different sensor systems: Landsat 5 Thematic Mapper (TM) for the epoch 2001–2012, Landsat 7 Enhanced Thematic Mapper Plus (ETM+) for the epoch 2001–2019 and Landsat 8 Operational Land Imager (OLI) for the epoch 2013–2019. Due to the failure of the Landsat-7 ETM+ Scan Line Corrector (SLC) in 2003, the ETM+ data are reduced by about 22% in each scene [42]. Rainfall estimates were derived from Climate Hazards group Infrared Precipitation with Stations (CHIRPS) data. The product is resampled to a spatial resolution of 0.05 degrees [43]. Following Funk et al. CHIRPS data have been largely used to examine rainfall trends and drought patterns in Ethiopia. The dataset is affected by uncertainties due to the inverse distance weighting function that is used for the blending procedure [43].

Polygon shapefiles for the micro-watersheds were provided by GIZ. Furthermore, between 2012 and 2018, the GFA Consulting Group collected georeferenced location data of soil and water conservation (SWC) measures that were implemented through SLMP to monitor the progress in the treated micro-watersheds. These data were used to relate vegetation development directly to SWC measure locations. The types of measures included in the dataset represent combinations of physical SWC constructions and biological activities (e.g., planting). Personal communication with SWC experts and project leaders during field visits in 2019 revealed that two types of measures included in the dataset, hillside terraces and check dams, should have a direct impact on the surrounding vegetation cover,

within a radius of approx. 500–1500 m. See Appendix A, Table A1 for details regarding the purpose and the number of geolocations available in the dataset.

### 2.3. Methods

All remote sensing data were acquired in Google Earth Engine (GEE), a cloud-based platform for geospatial data processing that stores a large repository of publicly available data [44]. Data processing and analysis were conducted using Python packages such as NumPy and Rasterio and the GEE client library (Figure 2).

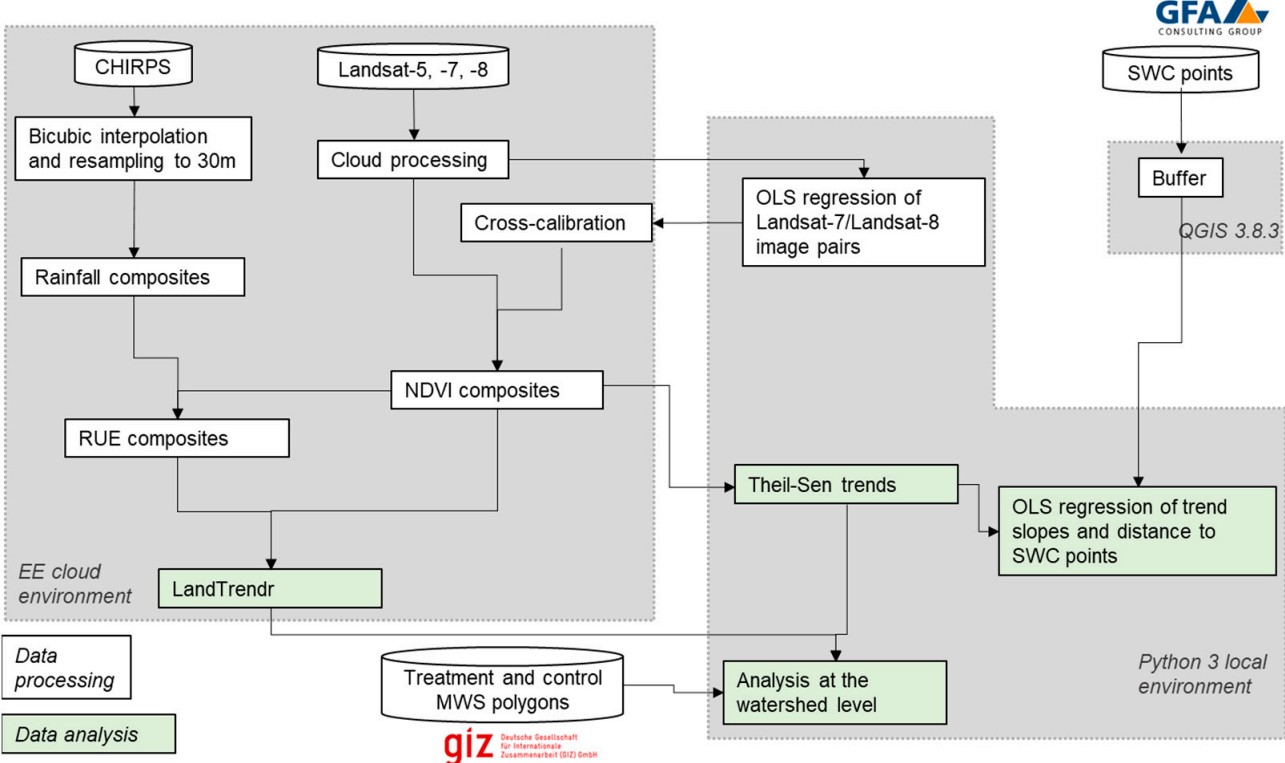

**Figure 2.** Workflow of the overall steps of the methodology.

### 2.3.1. Pre-Processing

For both cloud and cloud shadow masking, the C Function of Mask (CFMask), provided by USGS as pixel quality band as part of the Landsat products [45] was utilised. Visual inspection showed that using only CFMask proved to be insufficient. Therefore, clouds were additionally processed by the Google cloud score algorithm and cloud shadows by the Temporal Dark Outlier Mask (TDOM) [46]. Remote sensing analyses that integrate different sensor data require cross-calibration of the different datasets to ensure consistency [47–49]. As OLI spectral bands widths are narrower compared to ETM/ETM+, OLI NDVI values are on average higher [47] and it was therefore necessary to adjust OLI to ETM/ETM+ NDVI values before temporally aggregating the data. Transformation functions were developed using ordinary least squares (OLS) regression:

$$NDVIETM+ = a \times NDVIOLI + b \tag{1}$$

For this purpose, OLI and ETM+ NDVI images were paired based on the closest available dates. As OLI and ETM+ share the same orbit offset by 8 days, the western and eastern side of a sensor acquisition are overlapped by the eastern and western sides, respectively of the other sensor [47]. In these areas where the sensor acquisitions spatially overlap, image pairs are available with a temporal separation of only one day. To optimise

the inter-comparison, pixels used to produce the OLS models (Figure A1) were therefore extracted for the study area where overlap existed. To minimise problems related to changing surface states and conditions such as different crop cycle stages [47], pairing was done during the dry season (November to February), i.e., for two seasons, 2013/14 and 2017/18, for an area including a large range of vegetation densities (including evergreen vegetation).

NDVI has been used extensively for analysis of dryland vegetation [50] as NDVI saturation that can occur in densely vegetated areas is rarely a concern in drylands [51]. NDVI was here used as a proxy for the vegetation condition during the period of analysis. Annual NDVI maximum value composites (MVC) were produced based on all cloud-free available pixels during the period August through October, which represents in the end of the growing season. As image coverage was insufficient, a three-year (+1/−1 year) moving maximum window was used to fill data gaps (Figure A2). Annual rainfall composites were computed using the seasonal rainfall sum (March to September). As agriculture is primarily rainfed and crop productivity highly dependent on rainfall (MOFED, 2002) [52], the rain-use efficiency (RUE) was used as a proxy for assessing non-climate related changes in vegetation conditions [25,26] inherent to the implementation of the SLMP activities to counter land degradation. As vegetation productivity in the study area is predominantly determined by seasonal rainfall, RUE was calculated as the ratio of the maximum NDVI, as an approximation of ANPP, and the seasonal rainfall sum. For this purpose, the rainfall composites were resampled from 0.05 degrees to Landsat resolution of 30 m using bicubic interpolation.

### 2.3.2. Theil-Sen Regression and Mann-Kendall (MK) Trend Test

The temporal development of NDVI was used as an indicator of land degradation ('browning') and land recovery ('greening'). Spatiotemporal patterns of NDVI and rainfall were examined using the non-parametric Theil-Sen median slope [53] to analyse changes in vegetation condition (climate-related vs. human-induced). The Mann–Kendall (MK) test [54] was applied to evaluate NDVI trends at the 99% ($p < 0.01$) and rainfall trends at the 95% ($p < 0.05$) significance level. A stricter criterion ($p < 0.01$) for NDVI trends was applied due to the temporal smoothing of the Landsat-based NDVI time series. Pixel-wise slope differences were calculated between the two periods of 2002–2010 and 2011–2018. The split of the two periods is determined by the timing of the SLMP implementation and length of time series.

Aggregated NDVI trends were calculated for treatment and control areas using the median. This was done for the total study area ("regional" scale) as well as for each major watershed that comprises both treatment and control micro-watersheds ("local" scale). To assess the effects of treatments, i.e., SLMP interventions, the significance of the difference in the distribution of per-pixel-NDVI trends between the two sample groups was evaluated using the Mann-Whitney U test, which is a non-parametric test for independent samples, non-normally distributed data and different sample sizes [55]. Spatial patterns were assessed by inspecting trend maps with the additional use of multi-temporal Google Earth VHR images.

In order to investigate the spatiotemporal relationship between vegetation and rainfall, the agreement of NDVI and rainfall trend directions was computed on a pixel-basis for each period. This method is based on a model which, following Horion et al. [26], interprets the nature of changes in ecosystem functioning based on the combination of growing season vegetation and rainfall trends. A decrease in growing season vegetation despite an increase in precipitation or vice versa is likely to be caused by human activities. In contrast, trend combinations with the same direction of change are likely to be caused by climate (see Figure A3 for further explanation). The stronger the magnitudes of change in a given direction, the more likely the cause attribution [26].

### 2.3.3. LandTrendr

In addition to the trend analysis based on fixed time periods (Section 2.3.2), we conducted an analysis based on the LandTrendr approach in GEE [56]. This was done to further explore the importance of analysing land degradation and SLMP interventions at the level of Landsat pixel resolution rather than at coarser spatial resolution (e.g., MODIS predominantly being used for time series analysis) where subtle vegetation changes at local scale is likely to go unnoticed. The implementation of SLMP interventions was conducted at different times during the eight-year epoch 2011–2018 in the different micro-watersheds. Therefore, human-induced changes through SLMP occurred presumably within a time period shorter than eight years. Horion et al. argue that abrupt changes in RUE can indicate significant changes in ecosystem response to precipitation through human activities [26]. To identify trends with a shorter duration than eight years and to obtain more information about the types and timing of the changes, LandTrendr was applied to NDVI and RUE composites. The algorithm was used to fit a model for the period 2002–2018 with a maximum number of three segments (for the sake of simplicity) and a confidence interval of 95% ($p \leq 0.05$) determining the significance of the fitted segments. For each significant segment, the algorithm returns the magnitude, duration and rate of change as well as the start year in which the segment was detected, the end year and the corresponding NDVI value defined by the identified vertices (Figure A4). The Pearson's correlation coefficient (r) was used to mask pixels where RUE correlated with rainfall over the overall period 2002–2018 using a confidence interval of 95% ($p < 0.05$). This was done, as the use of satellite-based RUE time series to identify non-precipitation related land degradation/recovery is problematic for pixels where RUE remains correlated with NDVI, as this suggests NDVI changes still to be controlled by changes in precipitation [25,57,58].

### 2.3.4. Effect of Soil and Water Conservation (SWC) Measures on Vegetation Trends

OLS regression was used to estimate the effect of SWC measures on vegetation trends where the trend represents the dependent variable and the distance to the SWC points the explanatory variable. To investigate the influence of SWC distance on the trends, buffers were created based on the geolocations of the different SWC types and implemented based on three different sizes: buffers with a 250 m and 500 m radius, both with a zone width of 50 m (Figure A5), and buffers with a 1000 m radius and a zone width of 100 m. Within each zone the trend results were aggregated in two ways; first, by using the median of all Theil-Sen trends and second, by using the proportion of the significant Theil-Sen as well as LandTrendr increases. The aggregated results were regressed against the corresponding distance value of the zone.

## 3. Results

MK trend test revealed that rainfall trends were not significant in any of the sub-periods apart from 0.3% of the study area with increasing trends in 2002–2010 ($p < 0.05$). Over the entire study period 2002–2018, monotonic increasing trends were found for 17% and decreasing trends for 1.3% of the total area. When considering all rainfall trends, increasing trends were observed in 91% of the entire study area for the sub-period 2002–2010, 54% for the sub-period 2011–2018 and 78% for the entire study period 2002–2018.

The increasing rainfall trends during 2002–2010 had a median annual change of 15.2 mm and occurred with mainly decreasing NDVI trends (Figure 3a). During 2011–2018 rainfall experienced almost no or little changes (median annual change of 1.5 mm) while NDVI was predominately increasing (Figure 3b). Over the entire study period 2002–2018, minimal increases in rainfall with a median annual change of 4.2 mm occurred with increases in NDVI (Figure 3c). When considering only significant rainfall trends for this period (Figure 3d), the agreement shows a more pronounced pattern of positive significant monotonic NDVI and rainfall increases with an annual change rate of 15.8 mm per year.

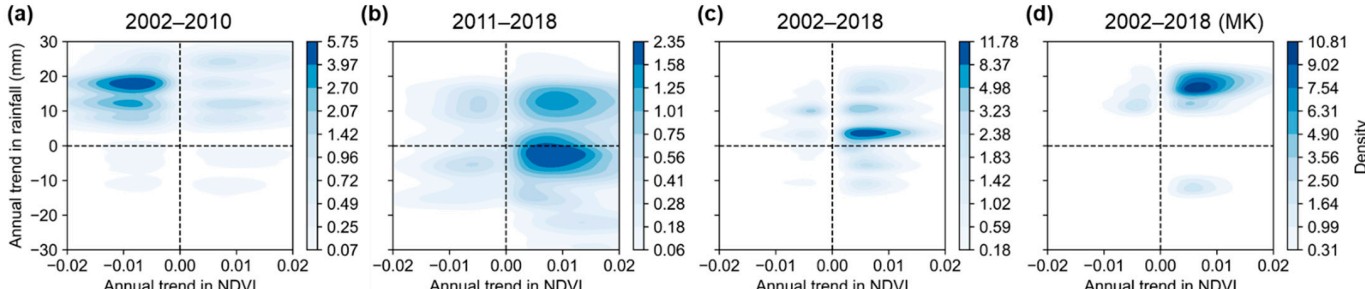

**Figure 3.** Spatial agreement of all annual rainfall trends and significant NDVI trends ($p < 0.01$) as density plots for the total study area for (**a**) the period 2002–2010, (**b**) 2011–2018 and (**c**) 2002–2018, and for (**d**) 2002–2018 with significant rainfall trends ($p < 0.05$).

### 3.1. Treatment and Control Areas

#### 3.1.1. Theil-Sen Trends

In the period 2002–2010 the median trend in NDVI was negative with an annual decrease of −0.0065, while in 2011–2018 NDVI annually increased by 0.009 (Figure 4a). The proportion of pixels where significant NDVI trends occurred in both periods accounted for 2.6% (treatment) and 2.8% (control). The distributions of the trend differences of these pixels are left-skewed for both treatment and control areas and have a median of 0.019 (treatment) and 0.018 (control) (Figure 4b).

The most common trend was a shift from negative to positive (Figure 4c). The proportion of significant negative-positive trends was slightly higher for treatment areas (1.84%) than for control areas (1.79%) (Figure 4c). The overall trends (all trend types included), the negative-positive trends, and trends with the same sign in both sub-periods all had a positive median trend (Figure 4d). The median trends were similar for both sample groups.

The results of the individual watersheds reveal that cases where treatment areas have larger trends than control areas dominated (Table A2). When treatment and control micro-watersheds from all major watersheds were treated as two large sample groups, the overall NDVI trend was larger for treatment (0.019) than for control (0.0183).

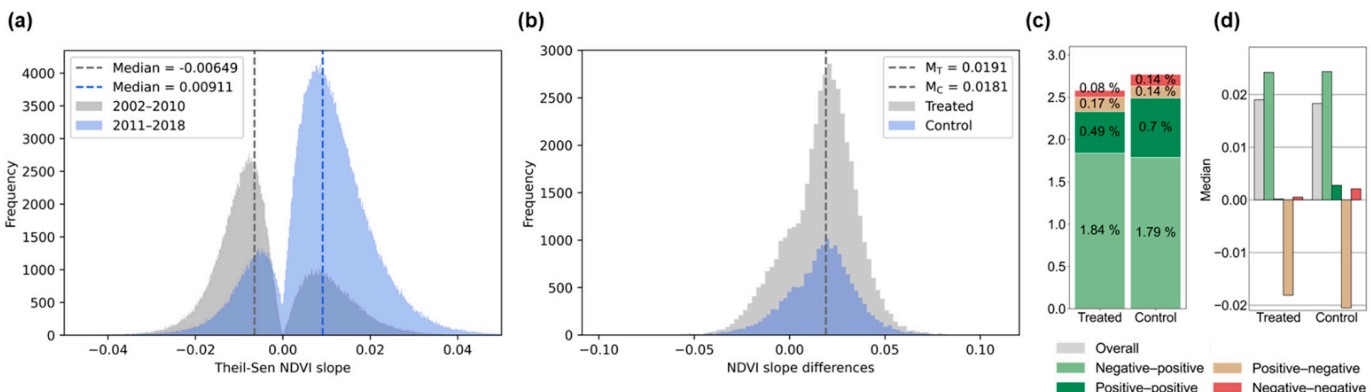

**Figure 4.** Theil-Sen NDVI trends. (**a**) Distribution of statistically significant ($p < 0.01$) trends for both sub-periods for the total study area; (**b**) Distribution of NDVI trend differences (slope of 2011–2018 minus slope of 2002–2010) for treatment and control areas; (**c**) Proportion of each type of change for all significant pixels and (**d**) Median of NDVI trend differences (slope of 2011–2018 minus slope of 2002–2010) for the overall trend and for each type of change constellation.

#### 3.1.2. LandTrendr

A LandTrendr-based approach was used to refine the Theil-Sen trend analysis that was based on two fixed time periods defined by the major scheme of SLMP support. LandTrendr detected for 46% of both treatment and control areas statistically significant

($p \leq 0.05$) changes in NDVI for the entire study period 2002–2018, that were subsequently analysed as a function of change types. For 92% of the study area pixels showed negative correlation of RUE and rainfall and were therefore not included in the change type analysis (Section 2.3.3). The remaining significant RUE changes (that were not correlated with rainfall changes) accounted for 2% of treatment and 3% of control areas.

The most frequent NDVI change type (amongst the 46% of statistically significant pixels) consisted of one decreasing followed by an increasing trend for 29 and 28% of all significant pixels in treatment and control areas, respectively (Figure 5a). The second most frequent change type were two consecutive increasing trends (24 and 26%, respectively). For RUE, the most common trends were of the same type as for NDVI, however with a larger share of two consecutive increasing trends (44 and 42%, respectively).

To examine the timing of detected trend segments for treatment and control areas, the onset of the decreasing and increasing trend segments with the greatest rate were extracted from each pixel (i.e., from each trend sequence) and aggregated in two groups, respectively. Generally, treatment and control areas showed the same pattern in the timing of trends without any pronounced differences. For both sample groups, the timing of the onset of approx. 90% of both NDVI and RUE decreasing trends was in 2002 (Figure 5b). In contrast, only approx. 40% of the NDVI increasing trends for both groups occurred in this year. For RUE increasing trends the proportions amounted to 53% (treatment) and 51% (control). The remaining NDVI and RUE recoveries occurred proportionally more equally distributed over the years after 2002 (2–10% in the remaining years of the time series).

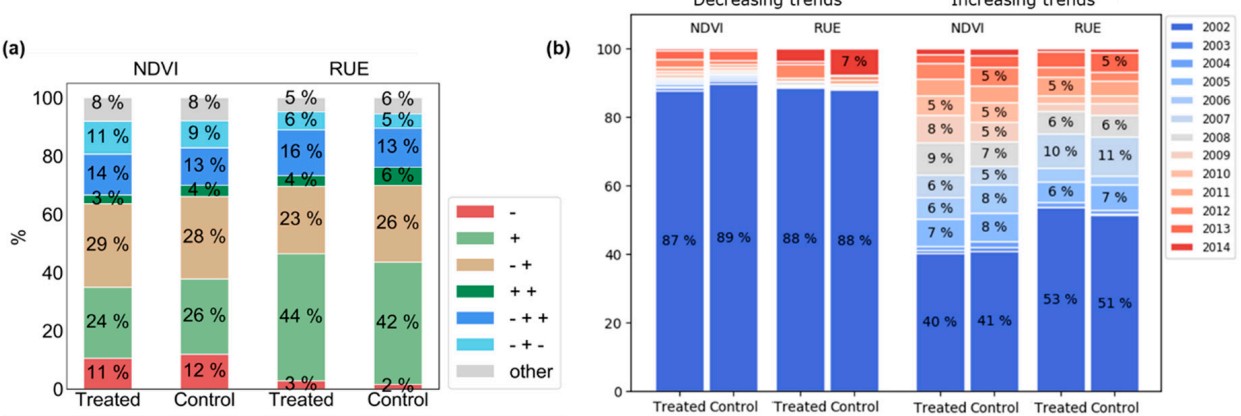

**Figure 5.** LandTrendr results for treatment and control areas, respectively; (**a**) Change types detected by LandTrendr with "+" indicating increasing and "-"decreasing trends. As the change types resulted from fitting either one, two or three segments into the time series, different combinations of trends existed. For example, while "-" indicates a decreasing trend over the entire study period, "- + +" indicates one decreasing trend followed by two increasing trends; (**b**) Timing of NDVI and RUE trends.

### 3.2. Visual Inspections of Trends Using Google Earth

Google Earth imagery was used to inspect all 21 major watersheds and showed that the limited amount of significant RUE trends to a wide extent occurred in areas where tree plantations have been established extensively. The watershed of Banja was selected as an exemplary case of illustration (Figure 6) to link interventions on the ground with the trends captured by the remote sensing time series data. Here, the positive trend differences represent pixels where trees have been planted between 2011 and 2018 and negative trend differences were mostly found in croplands, while grazing land showed no change in NDVI (Figure 6a). NDVI recoveries (an increasing trend segment) detected by LandTrendr were observed to be more dispersed in time (Figure 6c) and space (Figure 6d) as compared the RUE (Figure 6e). The higher uniformity of the RUE results at the pixel level can be explained by the coarser resolution of the CHIRPS rainfall data used to compute RUE time

series at the scale of Landsat NDVI data. The strongest changes in RUE were therefore more likely to be found in the same year for different pixels as compared to changes in NDVI. RUE recoveries were mostly detected between 2013 and 2014 (Figure 6c) and represent pixels where trees have been grown on former cropland or grazing land (Figure 6e). This is exemplified in the time series (Figure 6b) at the point of interest (POI) where LandTrendr detected the strongest NDVI recovery in 2012 that continued until 2017 with a rate of 0.07 per year. The rather high NDVI and low rainfall in 2015 led to high RUE in this year; hence the detection of a positive RUE trend in 2013 with a rate of 0.09 per year. The VHR images show that while agricultural fields and grazing lands were largely unterraced in 2005 (Figure 6f), benches of terraces were fully established in 2013 (Figure 6g). From this year on, tree cover expanded until 2020 (Figure 6h).

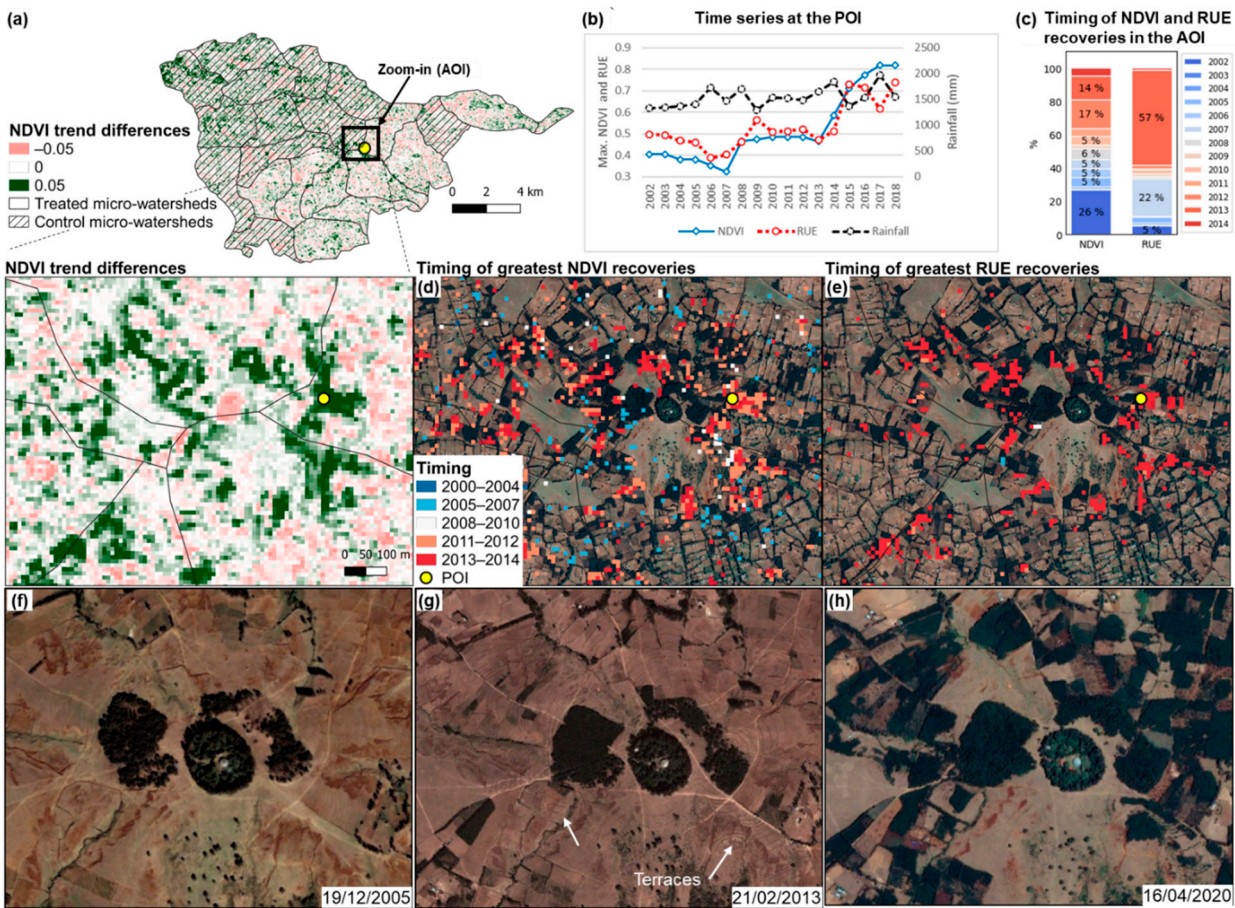

**Figure 6.** Watershed of Banja. (**a**) NDVI trend differences (slope of 2011–2018 minus slope of 2002–2010); (**b**) Time series at the POI; (**c**) Timing of the largest NDVI and RUE recoveries in the AOI; (**d**) Spatial distribution of the timing of the largest NDVI recoveries and of (**e**) RUE recoveries in the AOI; VHR images showing (**f**) mainly unterraced hillside in 2005, (**g**) the establishment of terraces in 2013 and (**h**) expanded tree cover area in 2020.

Furthermore, an inspection of Google Earth imagery from the watershed Yilmana Densa (Figure 7a) revealed that positive vegetation trends occurred mainly outside agricultural fields. Generally, the timing of NDVI recoveries in this watershed was equally distributed over the entire study period while RUE recoveries occurred mainly in 2009, 2011 and 2013 (Figure 7b). Recovery trends detected after 2010 were mainly detected along riverbanks, on steep hillsides, and on land affected by gully erosion. This is shown in area I (Figure 7c) and area II (Figure 7d) which accordingly showed positive trend differences that were associated with a trend shift of significant monotonic negative-positive trends. The two VHR images of the river in area I show that this development is related to a decline in eroded soil from 2013 to 2016. For area II, the VHR images show that while the hillside

was largely covered by degraded soil in 2014, it was revegetated in 2019 which explains the positive NDVI trends. Furthermore, area III (Figure 7e), characterised by degraded land and affected by gully formation, increased in vegetation cover between 2005 and 2019 which is reflected by the positive NDVI trend differences. This result also coincides with the evaluation through the performance assessment by GFA Consulting Group in which the area of the corresponding micro-watershed was described as "reclaimed land from gully erosion transformed into forage production and other economic activities" (personal communication with GFA's SLMP team leader).

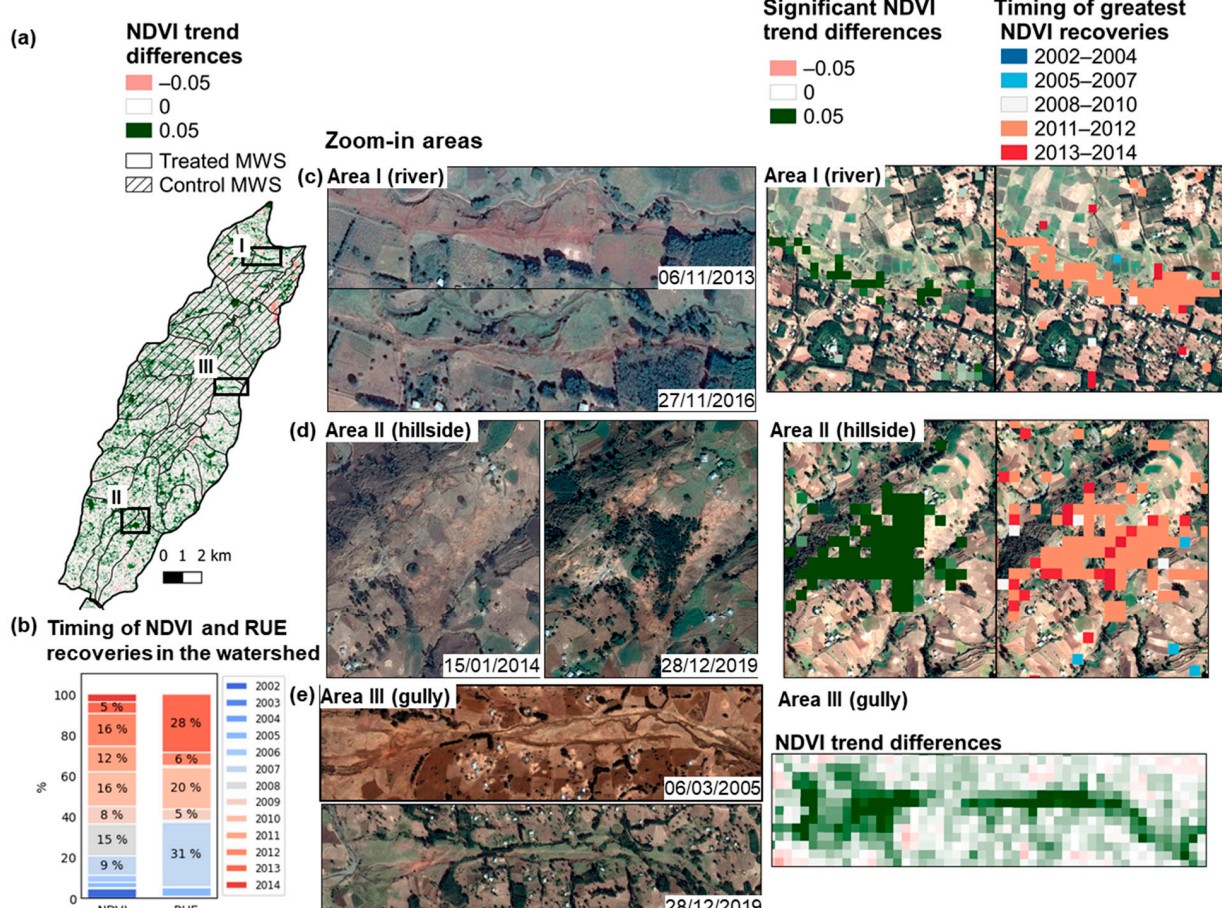

**Figure 7.** Watershed of Yilmana Densa. (**a**) NDVI trend differences (slope of 2011–2018 minus slope of 2002–2010); (**b**) Timing of the largest NDVI and RUE recoveries; Zoom-in areas showing (**c**) a decline of eroded soil at a riverbank (area I), (**d**) an increase in vegetation along a hillside (area II) and (**e**) an increase in vegetation in and nearby a gully (area III) with corresponding trend patterns.

### 3.3. Effect of SWC Measures on Vegetation Trends

The regional regression results (results based on the available SWC data in the entire study area) showed strong negative relationships of trends and distance particularly for check dams. The strongest relationships were found between the distance and the median trend differences (slope of 2011–2018 minus slope of 2002–2010) within 250 m (r = −0.97 **, $r^2$ = 0.94) and 500 m (r = −0.98 ***, $r^2$ = 0.8) (Figure A6a, Table 1) as well as between the distance and the proportion of significant increases 2011–2018 within 250 m (r = −0.97 **, $r^2$ = 0.93) and 500 m (r = −0.95 ***, $r^2$ = 0.91) (Figure A6b, Table 2). For the latter type of trends, terraces showed strong negative relationships within 500 m (r = 0.87 **, $r^2$ = 0.76) and 1500 m (r = 0.85 ***, $r^2$ = 0.73) radius (Table 2). Otherwise, linear relationships within 1500 m buffers were for both SWC types predominately weak.

**Table 1.** Regional OLS regression results for median trend differences (all trends).

| SWC Type | Buffer Option 1 (250 m) | | | Buffer Option 2 (500 m) | | | Buffer Option 3 (1500 m) | | |
|---|---|---|---|---|---|---|---|---|---|
| | Slope | r | r² | Slope | r | r² | Slope | r | r² |
| Check dams | $-1.7 \times 10^{-5}$ | $-0.97$ ** | 0.94 | $-8.60 \times 10^{-6}$ | $-0.98$ *** | 0.8 | $-4.00 \times 10^{-7}$ | 0.23 | 0.05 |
| Terraces | $-2.0 \times 10^{-7}$ | $-0.02$ | 0 | $-2.00 \times 10^{-7}$ | $-0.06$ | 0 | $-8.00 \times 10^{-7}$ | $-0.58$ * | 0.34 |

Differences exist with different significance levels (* $p < 0.05$, ** $p < 0.01$, *** $p < 0.001$).

**Table 2.** Regional OLS regression results for the proportion of significant positive trends in the period 2011–2018.

| SWC Type | Buffer Option 1 (250 m) | | | Buffer Option 2 (500 m) | | | Buffer Option 3 (1500 m) | | |
|---|---|---|---|---|---|---|---|---|---|
| | Slope | r | r² | Slope | r | r² | Slope | r | r² |
| Check dams | $-0.046$ | $-0.97$ ** | 0.93 | $-0.03$ | $-0.95$ *** | 0.91 | $-0.007$ | $-0.79$ *** | 0.63 |
| Terraces | $-0.035$ | $-0.64$ | 0.42 | $-0.029$ | $-0.87$ ** | 0.76 | $-0.010$ | $-0.85$ *** | 0.73 |

Differences exist with different significance levels (** $p < 0.01$, *** $p < 0.001$).

The results showed stronger correlations for a few local regressions models (results of the individual watersheds). An example is the watershed of Tahtay Koraro that experienced increasing trends particularly along the micro-watershed borders where steep slopes exist. Strong negative relationships of trends and the distance to both check dams and terraces were observed. Significant increases between 2011 and 2018 clustered at the location of two check dams and one terrace (Figure 8a). The strongest relationship was found within a 250 m radius ($r^2 = 0.98$) with a decreasing proportion of significant trends from 74% at 50 m, to 36% at 250 m and to 13% at 1500 m (Figure 8b). The VHR images reveal that the hillsides were, in accordance with the SWC data, terraced in 2016 and increased in vegetation cover up to 2019 (Figure 8c).

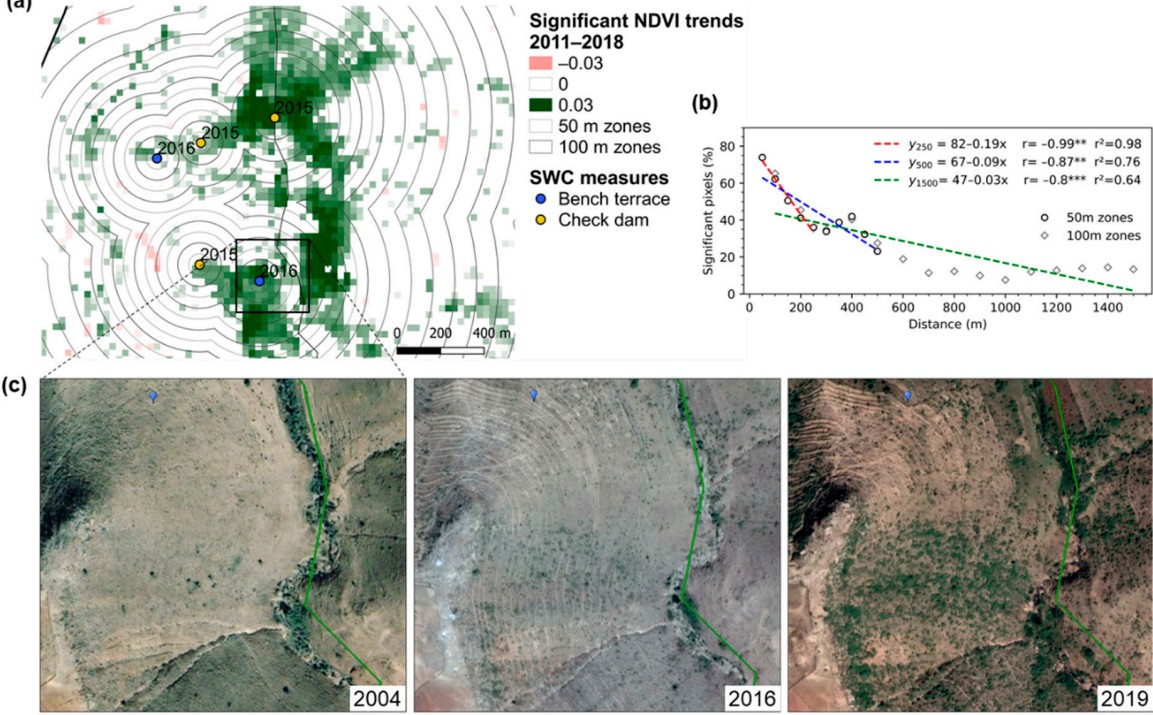

**Figure 8.** OLS regression of trends and the locations of SWC measures in the watershed of Tahtay Koraro. (**a**) Significant NDVI trends 2011–2018 with SWC points showing the year of completion; (**b**) Combined regression model for all check dams and terraces in the watershed for the proportion of significant increasing NDVI trends 2011–2018; (**c**) Multi-temporal Google Earth VHR images showing the zoom-in area with an increase in vegetation cover from 2004 to 2019 and the construction of terraces in 2016. ** $p < 0.01$, *** $p < 0.001$.

Another example of spatial correlations of increasing NDVI trends and SWC measure locations was observed for the watershed of Gudeyabila where the trend differences (Figure 9a) and recovery trends detected after 2010 (Figure 9b) clustered in close proximity to a check dam. Here, a decrease of the median NDVI from 0.015 at 50 m to 0.005 at 250 m ($r^2 = 0.85$) was observed (Figure 9c). The VHR images show the existence of a gully as well as mainly unterraced hillside in 2013 (Figure 9d). The construction of the check dam, which was completed at the gully in 2015, was accompanied by treatment through terraces of the surrounding hillside area including grazing land and cropland.

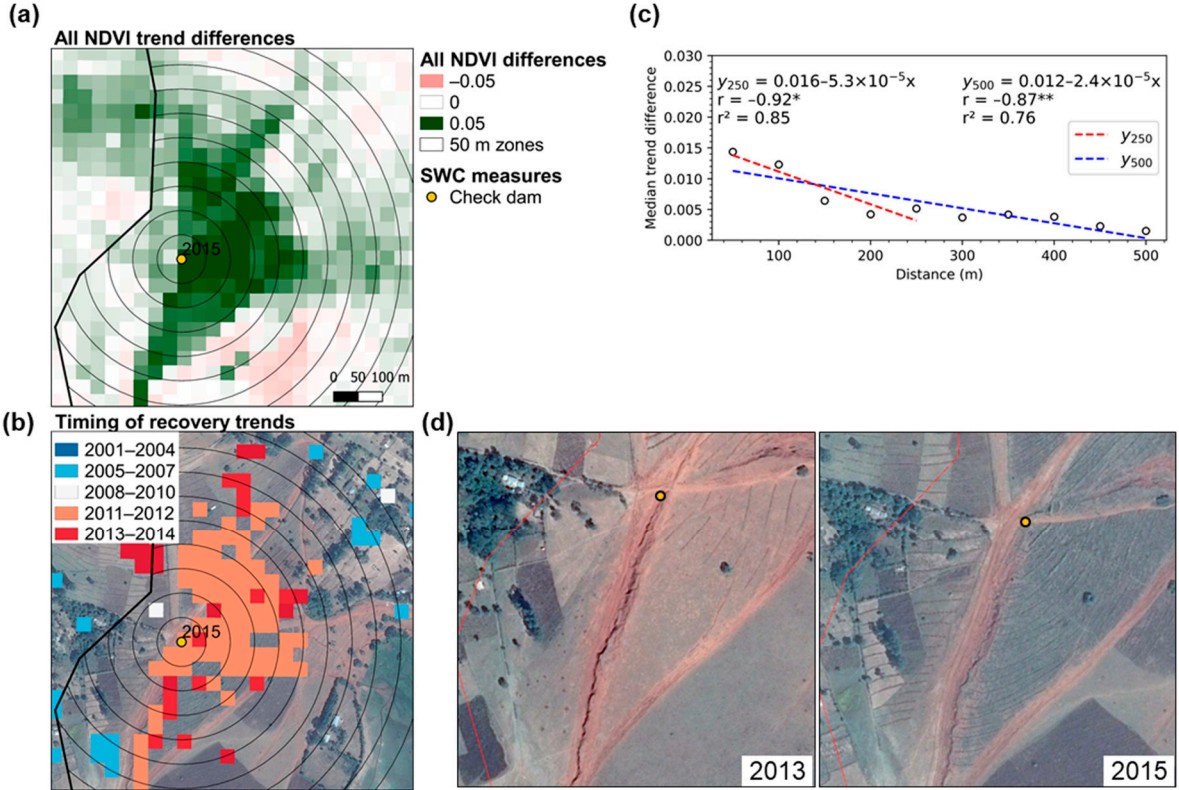

**Figure 9.** OLS regression of trends at a check dam location in the watershed of Gudeyabila; (**a**) Trend differences (slope of 2011–2018 minus slope of 2002–2010); (**b**) Timing of recovery trends; (**c**) Regression model of the median trend difference for the check dam location; (**d**) Google Earth VHR images showing mainly unterraced hillside in 2013 and the construction of terracing in 2015. * $p < 0.05$, ** $p < 0.01$.

## 4. Discussion

During the entire period 2002–2018 and during the second sub-period 2011–2018, the study area showed more pixels of NDVI increase than decrease and vice-versa during the sub-period 2002–2010 (Figures 3 and 4a). The browning trends in the latter period coincide with previous findings by Hermans-Neumann et al. who identified declining net primary production between 2000 and 2009 in the highlands of Amhara and Oromia [18]. Besides the greening trends between 2002 and 2018, several results of this study indicate a shift from browning to greening within the same period. On the one hand, this was indicated by the results from the Theil-Sen trend analysis with a shift from decreases to increases in 2011 (Figure 4c), on the other hand, browning to greening was also expressed by the LandTrendr results showing a relatively large number of the negative-positive change types (Figure 5a). For RUE, this change type was the second most common one. The general shift from negative to positive trends was also in agreement with the timing of increases and decreases, as decreases occurred mainly in 2002 and increases mainly after 2010 (Figure 5b).

14% of the study area experienced a significant increase in both NDVI and rainfall in the overall study period 2002–2018. This spatiotemporal pattern indicates climate-related greening [26]. In the first epoch 2002–2010, the study area presented mainly negative rates of change in NDVI despite increasing (but not significant) rainfall trends (Figure 3a). Even though this pattern indicates human-induced browning, evidence of this is not provided due to the insignificance of the rainfall trends and the results rather suggest that declining NDVI was not related to a long-term change in rainfall. Rainfall variability in North, West and central Ethiopia increased during the period 1983 to 2012 [59]. As extreme weather conditions pose major challenges to agricultural activities through reduced crop yields and intensified soil erosion [59], changing intra-annual patterns are overlooked by using seasonally summed rainfall and the results could possibly be improved by examining trends in rainfall intra-annual variability [60].

### 4.1. Treatment and Control Areas

The Mann–Whitney U test results showed larger median NDVI trends between 2011 and 2018 for treatment areas than for control areas at the regional level with most local test results (i.e., within each individual major watershed) indicating this as well (Table A2). However, the medians differed only marginally and the significant differences in the distributions of the per-pixel trends were not immediately apparent in the trend maps. The significance of test results may be attributed to large sample sizes facilitating distributions to be significant.

The LandTrendr results did not reveal any substantial differences in the types and timing of changes between treatment and control areas (Figure 5); hence did not provide evidence for an improved development of treatment than control areas. However, it should be considered that this result may be tied to the applied methodology, particularly the use of the maximum NDVI which may not fully capture effects of interventions. This may be the case if the impact of interventions show an increase in crop yield that might be reflected better as an increase in the integral of the phenological crop cycle curve (integrated NDVI) or if the impact is the reduction of eroded soil in the end of the rainy season which could lead to higher NDVI values in the end of the phenological cycle, rather than causing an increase in maximum NDVI. Future research should, provided that the data availability will be sufficient (e.g., Sentinel-2), therefore include seasonal metrics when examining changes in vegetation condition related to SLM interventions.

### 4.2. Visual Inspections of Trends Using Google Earth

Whereas the previously discussed results based on the analysis conducted and reported at the level of watersheds did not indicate strong differences in vegetation development between treatment and control areas, closer visual examination of the trend maps showed that human-induced land improvements can be detected from the Landsat-based approach developed, though at localised scales rather than consistently spread throughout entire watersheds. For a few major watersheds, particularly for Banja and Sekela in Amahara region, the VHR images supported that RUE increases between 2012 and 2014 were due to the establishment of tree plantations (Figure 6). Following Mekonnen et al., communal and private lands in rural areas in Amhara region have been extensively used for the expansion of Eucalyptus and Acia plantations due to the demand for wood resource [61]. Furthermore, the government has been promoting tree planting widely through the introduction of campaigns. Consequently, woodlots, home gardens, trees on cropland and farm boundary plantations have become common agroforestry practices [61] which may explain the general emergence of small tree patches in various watersheds.

Furthermore, from visual inspection it was observed that notable changes occurred outside cultivated fields as patches of increasing trends in vegetation cover, particularly at hillside locations and along streams and gullies (Figure 7). These clusters of pixels were characterised by a negative-positive trend shift (MK) and were in agreement with LandTrendr recoveries of which the largest were detected after 2010. Since in these instances

VHR images could verify the regrowth of vegetation, these negative-positive trend shifts can be linked to anthropogenic land improvement where degradation previously occurred. Moreover, findings of positive development could be confirmed by results from the GFA Consulting Group performance assessment (Section 3.2).

*4.3. Effect of SWC Measures on Vegetation Trends*

The OLS regression results demonstrated that positive changes in NDVI could be attributed to the impact of SLMP infrastructure. Visual inspection showed that this was visible in the spatial patterns for several locations at check dams and terraces, as shown in the case examples (Figures 8 and 9). This coincides with results from a study by Ali et al. who used NDVI and other satellite derived indices to evaluate the impact of SWC measures on different land use (i.e., on cultivated land and non-cultivated land such as degraded hillsides), and found that biophysical measures had a particular high impact on non-cultivated land [13].

In general, the most significant positive changes in NDVI were observed within the smallest buffer zone; i.e., the decrease in the medium change rate or density of change occurrence was mainly observed within 250 m distance from the location, in few cases within 500 m (Tables 1 and 2). At distances larger than this and up to 1500 m, trends levelled off or even increased again. Field visits showed that gullies typically pass through cultivated fields and in these cases revegetation efforts are conducted mainly directly at the gully only. Hence, for land restoration of gullies dense vegetation regrowth does not often take place across larger distances from the restoration activities. Land improvement in proximity to check dams could therefore also occur as a line type pattern in the trend map. In this case, circular buffer zones will not help explaining vegetation trends. Apart from this, it should be considered that rehabilitation activities can occur at a smaller scale than Landsat's spatial resolution of 30 × 30 m, in which cases changes would not easily be detected. The spatial scale of the impact of SWC interventions as implemented in Ethiopia therefore underlines the challenge of detecting changes related to improved land management in developing countries based on the use of traditional remote sensing methods for change detection.

## 5. Conclusions

The aim of this study was to examine vegetation dynamics between 2002 to 2018 in degraded areas in the Ethiopian highlands and assess the impact of SLMP interventions. To examine vegetation dynamics in complex landscapes in Ethiopia on a detailed spatial scale, we investigated the potential of combining remote sensing data from different Landsat sensors using cloud-based geospatial processing supporting a high-resolution time series analysis. The vegetation dynamics in the study areas showed a shift from browning (2002–2010) to greening (2011–2018) along with an overall greening trend over the full period (2002–2018). From the spatiotemporal patterns of NDVI and rainfall it could be concluded that the browning trend was not explained by long-term changes in rainfall. In contrast, the greening trend over the full period could—for 14% of the study area—be explained by increases in rainfall. Overall, no clear patterns of anthropogenic induced changes in vegetation were found when aggregating results at the catchment scale, as NDVI median trends did not clearly indicate better development in SLMP intervention areas than in control areas. Visual inspection based on multi-temporal Google Earth imagery showed that the changes in NDVI and rain-use efficiency did spatially overlap areas of small-scale land improvements related to human management, however, on a smaller scale than a micro-watershed (the smallest aggregation level). The OLS regression results provided evidence of land recovery that could be attributed particularly to SLMP infrastructure (check dams and terraces). Positive impacts on vegetation were found to be contributing to improving the rehabilitation of degraded hillside areas and gullies. These findings underline that the little differences found between treatment and control areas when aggregated to the level of (micro-)watersheds are rooted in a scale issue, and

highlight the need for per-pixel trend analysis using sensor systems like Landsat, or higher spatial resolution, to be able to remotely capture the effect of SLMP interventions.

The ecological improvements through SLMP, identified here at the per-pixel level from the use of Landsat time series, are an important contribution to restore terrestrial ecosystems as targeted in the Sustainable Development Goals. Continuous efforts in developing means for improved monitoring of human-induced vegetation restoration of degraded lands will be essential to maintain rehabilitated land, prevent further land degradation and support environmental sustainability.

**Author Contributions:** Conceptualization, E.B. and R.F.; methodology, E.B. and R.F.; formal analysis, E.B.; writing—original draft preparation, E.B.; writing—review and editing, E.B. and R.F.; visualization, E.B. All authors have read and agreed to the published version of the manuscript.

**Funding:** R.F. acknowledges support by the Villum Foundation through the project 'Deep Learning and Remote Sensing for Unlocking Global Ecosystem Resource Dynamics' (DeReEco).

**Institutional Review Board Statement:** Not applicable.

**Informed Consent Statement:** Not applicable.

**Data Availability Statement:** The data presented in this study are available on request from the corresponding author.

**Acknowledgments:** The authors would like to thank GFA Consulting Group GmbH who provided the geolocation data on soil and water conservation measures and the German Agency for International Cooporation (GIZ) GmbH who provided the data on treatment and control areas.

**Conflicts of Interest:** The authors declare no conflict of interest. The funders had no role in the design of the study; in the collection, analyses, or interpretation of data; in the writing of the manuscript, or in the decision to publish the results.

## Appendix A

**Table A1.** Description of soil and water conservation (SWC) measure data.

| Type of SWC Measure | Purpose | Number of Geolocation Points |
| --- | --- | --- |
| Hillside terraces | Terraces are built to stabilise cultivated land, or to stabilise area. | 11 |
| Check dams | Check dams are obstruction walls constructed at the bottom of a gully, small streams or trenches in order to reduce run-off volume and prevent further widening of the gully channel [62]. These treatment measures are typically combined with revegetation activities to gain higher run-off infiltration into the sediments. | 43 |

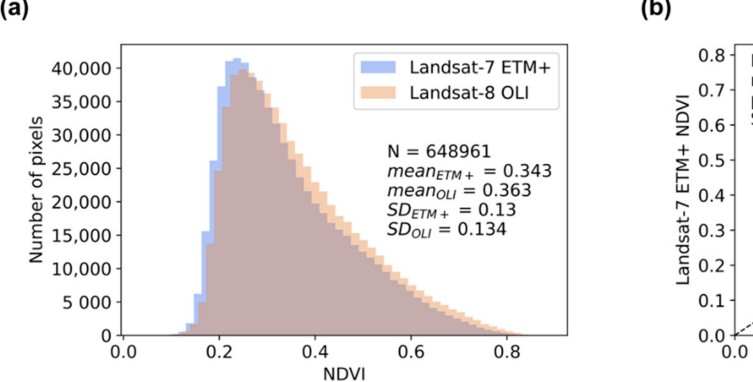
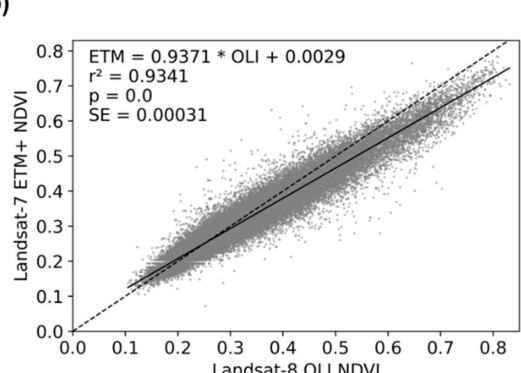

**Figure A1.** Landsat cross-calibration. (**a**) Distributions of ETM+ and OLI NDVI values; (**b**) OLS regression of ETM+ NDVI against OLI NDVI values. For a given OLI NDVI value, the corresponding ETM+ value is usually lower.

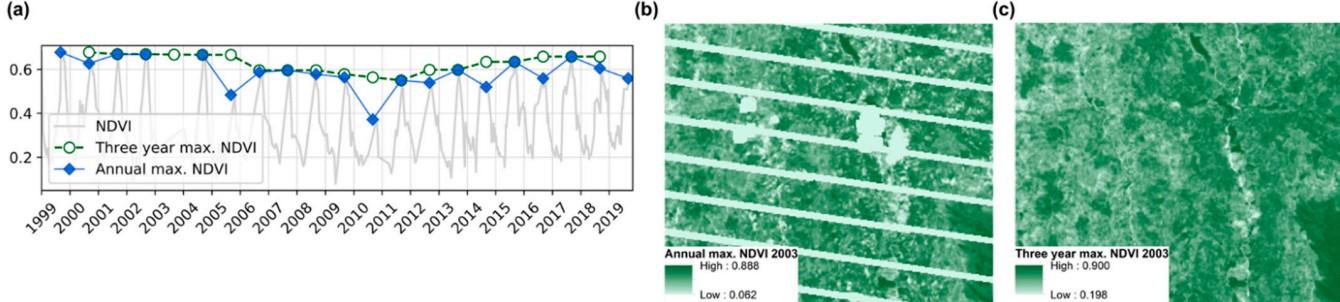

**Figure A2.** Interpolation. (**a**) Interpolation of a pixel's annual maximum NDVI time series (blue) through a three-year maximum moving window (green). The data gap in 2003 is interpolated; (**b**) Annual maximum value composite (MVC) with gaps produced by Landsat-7 ETM+ SLC error and clouds; (**c**) Interpolated MVC through applying a three-year maximum moving window.

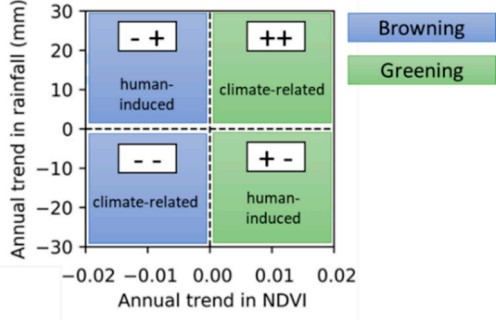

**Figure A3.** Interpretation of the nature of changes in ecosystem functioning based on the spatial agreement of vegetation (NDVI) and rainfall trends. A decrease in NDVI despite an increase in precipitation or vice versa is likely to be caused by human management. Trend combinations with the same slope direction are likely to be caused by climate change [26].

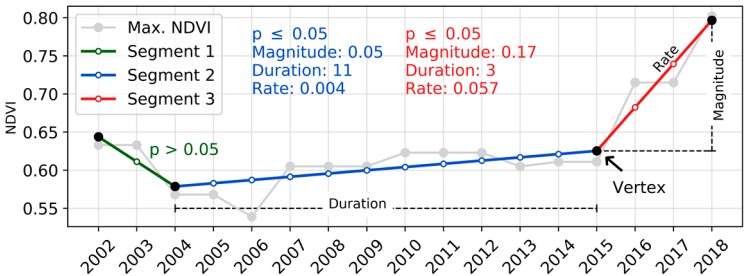

**Figure A4.** LandTrendr. Fitted segments into an annual maximum NDVI time series.

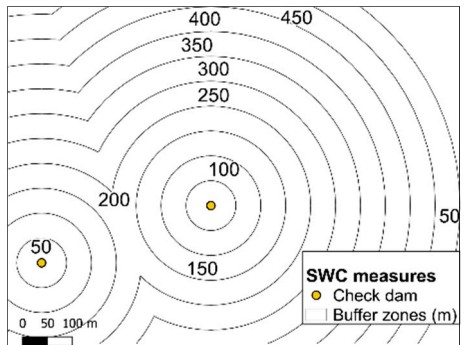

**Figure A5.** Example of buffer zones used for the OLS regression analysis to examine trends at soil and water conservation measure locations.

**Table A2.** Mann-Whitney U results with the sample size N (number of significant pixels), the median NDVI trend, and U indicating whether treatment areas have significantly [1] larger or smaller trends than control areas. Watersheds that did not include any control micro-watersheds are not included.

| Major Watershed | Treatment | | Control | | U |
|---|---|---|---|---|---|
| | N | Median | N | Median | |
| Laelay Adyabbo | 2091 | 0.0128 | 750 | 0.0116 | - |
| Tahtay Koraro | 6627 | 0.0203 | 604 | 0.0165 | Larger *** |
| Emba Alaje | 1080 | 0.0089 | 34 | −0.0091 | Larger *** |
| Gondar Zuriya | 693 | 0.0137 | 491 | 0.0172 | Smaller ** |
| Takusa | 2182 | 0.0208 | 2526 | 0.0130 | Larger *** |
| West Estie | 5248 | 0.0252 | 1203 | 0.0237 | Larger *** |
| Hagere Mariam | 1060 | 0.0065 | 60 | 0.0164 | Smaller *** |
| Sinan | 2135 | 0.0213 | 1324 | 0.0218 | - |
| Aneded | 3517 | 0.0292 | 2387 | 0.0270 | Larger *** |
| Yilmana Densa | 2539 | 0.0242 | 2204 | 0.0193 | Larger *** |
| Sekela | 2384 | 0.0212 | 972 | 0.0196 | Larger ** |
| Quarit | 2000 | 0.0222 | 1220 | 0.0183 | Larger *** |
| Banja | 2370 | −0.0003 | 2974 | 0.0052 | Smaller *** |
| Ale | 861 | 0.0112 | 131 | 0.0117 | - |
| All watersheds | 43,984 | 0.0190 | 16,880 | 0.0183 | Larger * |

[1] Differences exist with different significance levels (* $p < 0.05$, ** $p < 0.01$, *** $p < 0.001$).

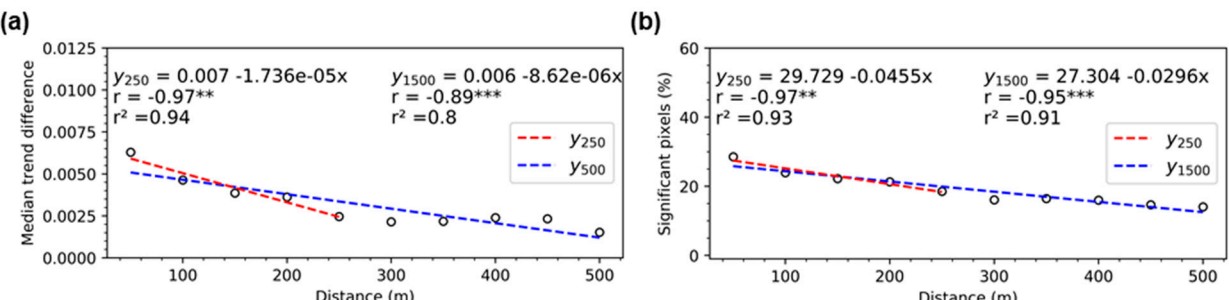

**Figure A6.** Regional OLS regression models for check dams with (**a**) the median of all trend differences and (**b**) the proportion of significant positive trends during 2011–2018 within the zones as dependent variable. The red line shows the linear slope when including zones up to 250 m distance, the blue line when including zones up to 500 m. ** $p < 0.01$, *** $p < 0.001$.

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
