# Peer review of "Earth Observation-Based Detectability of the Effects of Land Management Programmes to Counter Land Degradation: A Case Study from the Highlands of the Ethiopian Plateau"

_remotesensing, doi:10.3390/rs13071297_

Round 1

Reviewer 1 Report

This is a nice case study from the Ethiopian highlands, the results of which can potentially be helpful in informing land restoration elsewhere in the country as well.

My only comments are related to the use of NDVI and RUE, and the regional/watershed scale aggregations:

  • Did you consider other indices than NDVI for this analysis? This might be interesting given the limitations of NDVI in drylands where vegetation cover is predominanlty senescent and in forested ecosystems where this index tends to saturate.
  • Considering the rather coarse scale of the CHIRPS data used for calculating RUE, how do you explain the somewhat different patterns between NDVI and RUE recoveries (e.g. Figure 7)? It would be useful if you could elaborate more on this in the paper.
  • It is not entirely clear how you aggregated the data (NDVI and RUE) for the watersheds. Was this a simple mean/median or similar?

Reviewer 2 Report

This is an interesting, straightforward, and well written manuscript focused on an understudied and important part of the world. The authors consider temporal trends in NDVI in northern Ethiopia and how different management approaches may have influenced land cover over time. In general the figures are very nicely done and I especially appreciate the explanatory figures in the appendix. My overall comment is that there are a LOT of analyses in this paper that all say somewhat similar things, or seem to, and I would recommend that the authors consider either dropping some of the analyses or making a clearer argument for what each one brings (T-S regression vs LandTrendr, for example). Once it is established that there isn't really a trend in precip over the landscape, many of the analyses seem to keep asking this question anyway...

A few more broad comments:
(1) In several places (especially in the introduction and methods) the text reads more like a report to the KfW about SLMP than it does a scientific study. I think linking this more broadly to SDGs or other international development efforts would broaden its impact.

(2) Methods: a bit more should be said about how resampling the CHIRPS data to 30 m might impact the results. These data are provided at coarser resolutions because of data limitations, so resampling them could give readers a false sense of accuracy of these results. Similarly, why NDVI and not EVI or SAVI or some other index more appropriate for drylands?

Also, there is no explanation/discussion of the GE high resolution image analysis in the Methods. How and why were the example watersheds chosen? 

Line 271 - I am having a hard time following this sentence.

Line 278 - similarly I find this to be a bit hard to follow. What is the 'distance to the SWC point geometries'? 

Figure 3 - explain that this is plotted for each point, but as a density map (I think)? And are most of these not significant? Either I don't understand what these figures are showing, or why isn't 3d a subset of 3c?

Figure 4 (and results relating) - It looks like these are significant, but some are very nearly zero...?

Figure 5 - as with 3, I don't think the non-significant trends are very informative.

Figure 7 - note that many people are red/green colorblind, and so using a red-green color scheme will be difficult for those folks to see.

Figure 8 - I'm not totally sure what the farthest right (unlabelled) panels are showing? 

Discussion & Conclusions - I think it would be helpful if there were more references to figures/tables in the discussion. There are a lot of different analyses in this manuscript, and so it is hard to keep track or know where to look (for example, line 480 should refer to Figure 3, I think). It would also be good to be careful with language - for example, at line 452 the manuscript states "An overall monotonic increase in NDVI was observed..." but I don't think this was ever specifically tested (just the mean NDVI across the whole study area and time period)? So if really it is that a majority of pixels had a significant positive trend in NDVI, that is what should be stated (as I think is indicated in figure 3d, though I can't tell if this represents a majority or not). 

Lastly, the Discussion and Conclusions make contradictory statements about the impact of the SLMP infrastructure. I really like the zoomed in explorations in Figures 9 and 10, but these are anecdotal relative to the larger patterns. The Conclusions do point out that this is a scale issue, but I think this should be stated more clearly so that readers do not miss this distinction. 
